# A retrospective characterization of pediatric facemasks marketed in the United States and implications for future designs

Ali Hasani[1], Bryan Ibarra[1], Kirstie Snodderly[2], Dana Rottach[3], BiFeng Qian[3], Daniel Porter[1], Suvajyoti Guha[1]*

1 Division of Applied Mechanics, Office of Science and Engineering Laboratories, U.S. Food and Drug Administration, Silver Spring, Maryland, United States of America, 2 Office of the Chief Scientist, Office of Commissioner, U.S. Food and Drug Administration, Silver Spring, Maryland, United States of America, 3 Office of Product Evaluation and Quality, U.S. Food and Drug Administration, Silver Spring, Maryland, United States of America

* Suvajyoti.Guha@fda.hhs.gov

## Abstract

### Background

Device manufacturers who seek to market their pediatric facemasks in the United States (U.S.), as part of anthropometric data requirement, need to demonstrate their mask designs are expected to fit the intended user population. However, currently there are no well accepted test methodologies for pediatric facemasks. In addition, unlike N95 respirators, the expected maximum flow rate, and the pressure drop at that expected maximum flow rate for pediatric facemasks have not been established.

### Method

The objective of this article is three-fold; use a literature survey to determine the worst-case flow rate, and an acceptable breathing resistance; and come up with a bench-test based protocol for assessing fit of pediatric facemasks.

### Results & discussion

The worst-case breathing flow rate for mask testing in the pediatric population is 45–60 Liters/minute (LPM), and the acceptable pressure drop at the worst-case flow rate is 2.0 mmH$_2$O. A retrospective assessment of all the four brands of legally marketed facemasks in the U.S. that could be purchased, revealed that majority of the brands have high filtration efficiency (>95%) at low flow rate 5 LPM which reduces to ~ 80% at 45 LPM. At 5 LPM, the pressure drop ranges from 0.3–0.6 mmH$_2$O, remaining below the 2.0 mmH$_2$O. However, at higher flow rates, (representing strenuous activities, or older children (> 12 years)), most masks exhibited a pressure drop within the range of 2.9 to 6.0 mmH$_2$O. Furthermore, opening the pleats of the facemasks completely results in a notable reduction in pressure drop (a 6.6-fold decrease, p = 0.03). To assess fit of these same brands of facemasks, we then updated our previous validated adult manikin fit-test method and used it in manikins in the

**Data Availability Statement:** The data is available at https://figshare.com/s/52a88ceebdd563ee20aa.

**Funding:** COVID-19 Research Funding.

**Competing interests:** None to report.

age group of 2 to 14 years. Either poor nose-clip adherence to the manikin, low filtration efficiency of the pediatric facemasks, or off-label use (i.e. when donned on manikins representing 2 years to 14 years) contributed to low fit.

## Conclusions

A new bench-top tool to evaluate quantitative fit of pediatric facemasks was developed. In addition, based on the research reported here, we provide practical implications for the members of the community: users, academia and medical device manufacturers.

## Introduction

The Centers for Disease Control and Prevention (CDC) recommends that children aged 2 years and older wear masks to protect themselves and others from COVID-19, flu, and other illnesses. [1]. The technological requirements for pediatric masks differ significantly from those of N95 respirators, which are intended for use by healthy adults after a medical screening by a licensed medical professional (per 29 Code of Federal Regulations 1910.134(e)(2)). Due to children's unique anatomical and physiological characteristics, specialized masks tailored to their age group are necessary [2]. Smaller-sized masks intended for adults are available, but they were not designed to consider the risk of asphyxiation in children, and hence are not recommended for pediatric use. Pediatric facemasks, due to their intended respiratory protection use, are expected to ensure a proper fit and have resistance to passage of air, allowing children to breathe comfortably while remaining well protected [3]. Despite extensive research on N95 respirators, there has been limited focus on pediatric facemasks; currently there are no N95 respirators that are available for children. However, considering the widespread usage of masks during the COVID-19 pandemic, conducting additional research is crucial, especially regarding the fit and breathability of pediatric facemasks.

Although several pediatric facemasks have been US FDA-cleared to date to be legally marketed within the US (using product code OXZ in [4]), unlike N95 respirators for which 85 LPM is a well-accepted maximum flow rate for filtration efficiency testing [5], there is no consensus on the maximum flow rate for testing filtration efficiency of pediatric facemasks. In lieu of any consensus, device manufacturers use the surgical facemask standard, wherein the face velocity specified varies widely, ranging from 0.5–25 cm/sec. In addition, this standard is not specifically tailored to account for the unique physiological conditions of children [6]. Thus, *one objective* of this study is *to determine the suitable flow rate for testing masks intended for the pediatric population*.

Furthermore, the absence of established guidelines for acceptable pressure drop in pediatric facemasks adds another layer of complexity. For N95 respirators, the pressure drop limit is again well established [7] at 35 mmH$_2$O for inhalation at 85 Liters/minute (LPM). However, studies have found that high breathing resistance ($> 9$ mmH$_2$O) often causes discomfort in users [8]. Hence, modern day respirators are typically designed to have a breathing resistance of 6–9 mmH$_2$O. Unfortunately, there are no existing resources or references that provide information on the maximum permissible breathing resistance for pediatric facemasks. Thus, the *second objective* of this study was to draw insights from data related to adult N95 respirators to *define an acceptable pressure drop for pediatric facemasks* [8].

In evaluating the fit of pediatric facemasks, it's imperative to consider anthropometric requirements specific to the pediatric population as that is a FDA requirement for obtaining

marketing clearance of pediatric facemasks in the US [9]. However, to date, no methodologies have been developed to assess fit of pediatric facemasks and as a result, manufacturers have to often develop their own methods, which can often be inefficient and cost ineffective. Thus, a *third objective* of our study was *to develop a method for assessing fit and breathing resistance of future pediatric facemask designs.*

## Materials and methods

### Pediatric facemask models

At the time of execution of this study, there were nine brands of pediatric facemasks with 510 (k) clearance [4]. However, out of these nine, only four were available in the US market for purchase and those were selected for investigation in this study. In the text, we have presented the results in the order of their clearance, providing a chronological perspective on the development and performance of these pediatric facemasks. All of these masks are indicated for one-time use in children in the age group of 4 to 12 years and recommended for use in a health care setting with appropriate adult supervision.

### Worst-case flow rate for testing

While conducting tests on pediatric facemasks, determining an appropriate flow rate is crucial. The study commenced with a literature review, centering on the identification of suitable breathing flow rates for the pediatric population (S1 Text in S1 File).

### Maximum permissible pressure drop

Our search with terms like 'pediatric + breathing resistance' revealed a considerable variation in reported inspiratory resistance within this age group of 2 to 14 years, spanning from a minimum of 0.37 to a maximum of 2.1 $mmH_2O$/LPM [10–13]. We were not able to locate any direct references which define an acceptable pressure drop for pediatric facemasks. Although a standard for general barrier face coverings does mention a maximum pressure drop of 5 $mmH_2O$ for higher performance barrier face coverings, it does not specify any acceptable values for children. Hence, we relied on extrapolating adult response for N95 respirators to come up with the maximum permissible pressure drop in pediatric facemasks.

In adults, a higher tested pressure drop in respirators is reflected in increased breathing resistance, leading to discomfort and difficulty in breathing. To alleviate these symptoms, it is recommended to select respirators with a pressure drop below a certain threshold, typically in the range of 6–9 $mmH_2O$ [8, 14]. We assume that if the ratio of the resistance of pediatric facemasks by the total pressure drop in a pediatric lung maintained the same ratio as that of adults' respirators compared to total pressure drop in adults, then we can deduce the maximum permissible pressure drop in pediatric masks as follows–

$$\frac{(Maximum\ permissible\ pressure\ drop\ in\ pediatric\ facemasks)_{at\ highest\ flow\ rate\ for\ children}}{Total\ pressure\ drop\ due\ to\ pediatric\ lungs\ resistance} = \frac{(Maximum\ permissible\ pressure\ drop\ in\ adult\ respirators)_{at\ highest\ flow\ rate\ for\ adults}}{Total\ pressure\ drop\ due\ to\ adult\ lungs\ resistance} \quad (1)$$

Reorganizing Eq (1) above we get,

$$(\textit{Maximum permissible pressure drop in pediatric facemasks})_{\text{at highest flow rate for children}}$$

$$= \frac{(\textit{Maximum permissible pressure drop in adult respirators})_{\text{at highest flow rate for adults}}}{\textit{Total pressure drop due to adult lungs resistance}}$$

$$\times \textit{Total pressure drop due to pediatric lungs resistance} \tag{2}$$

The right-hand side (RHS) of Eq (2) can be determined using literature (S5 Table in S1 File) which then helps determine the left-hand side of the equation i.e. the maximum permissible pressure drop in pediatric masks (S2 Text in S1 File).

## Measuring filtration efficiency and pressure drop

We utilized our previous method for assessing filtration efficiency (Fig 1a) [3, 15, 16] under ideal conditions that are without any leaks. However, for pressure drop measurements, we explored three distinct methods: 1) using whole masks with pleats (Fig 2a), 2) using whole masks with pleats opened (Fig 2b), and 3) measuring pressure drop using facemask coupons. The first two methods were conducted utilizing the experimental setup illustrated in Fig 1a. To

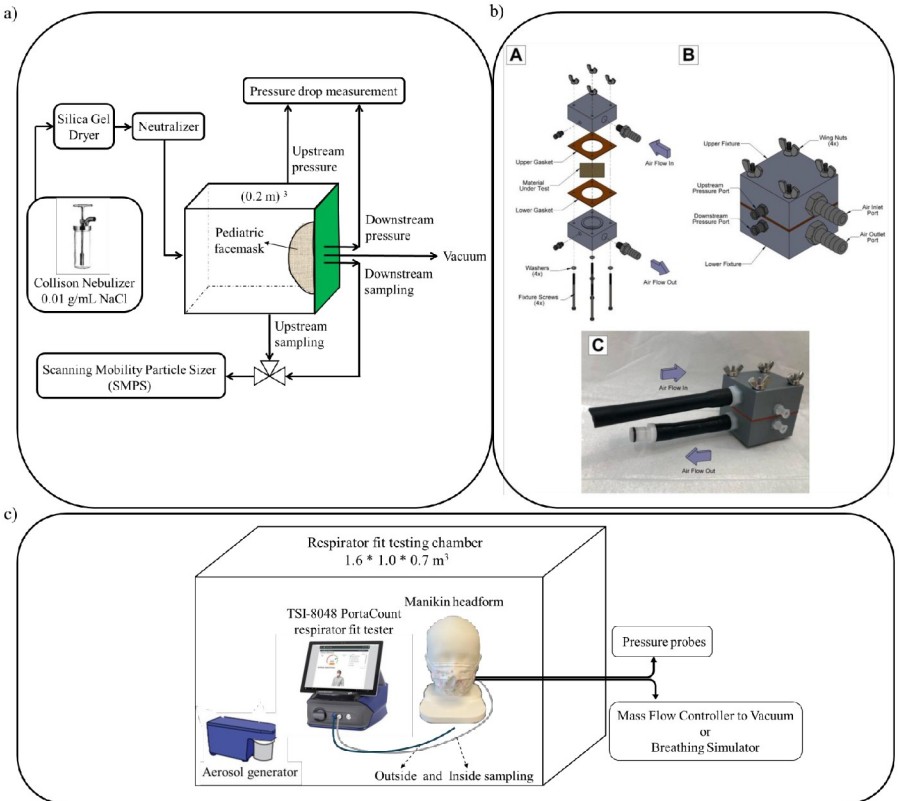

**Fig 1. The schematic representation of our experimental setups: a) for conducting filtration efficiency tests and measuring pressure drop, b) for measuring pressure drop across facemasks coupons, using a specialized 3D printed rig; [A] Exploded view of the 3D printed rig for measuring pressure, [B] Computer aided design of the rig, [C] Final 3D printed rig with tubings (adapted from reference [17]), c) for fit testing and measuring breathing resistance.** One of the pediatric headforms is shown here donning a pediatric facemask. Pressure probes were used to measure the breathing resistance both upstream and downstream of the facemask. All measurements shown below were made in triplicates.

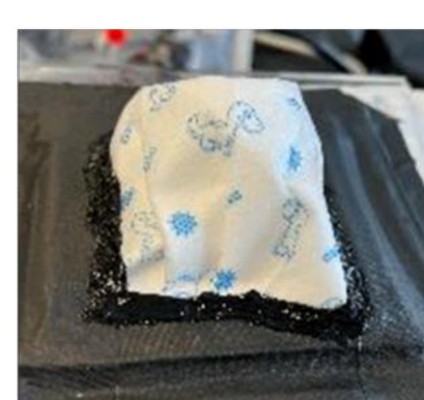

**Fig 2. Pediatric facemask brand C installed on plate for pressure drop measurements: a) whole mask with pleats, b) whole mask with opened up pleats.**

assess pressure drop in the coupons, we considered a coupon size of 4.9 cm$^2$ and utilized a method outlined in the literature, British National Standards (BSEN) 14683:2019 (Fig 1b), derived from MIL-M-36954C, and subsequently adapted here [17]. These coupons were directly cut from the whole mask, with all the inner layers retained (as in a whole mask), unpleated and then recut so the cross-sectional area would be 4.9 cm$^2$.

## Method for measuring fit and breathing resistance

Our fit measurement method was adapted from our prior research on adult manikins [18]. Fit testing and breathing resistance measurements were performed on additively manufactured headforms that featured a 5 mm imitation skin layer (Fig 1c). The selected headforms (S4 Text in S1 File) represented pediatric individuals aged 2 to 14 years, namely 2-year-old Betty, 5-year-old Roberta, 8-year-old Dizzy, 11-year-old Billie, and 14-year-old Louis [19] as shown in S6-S8 Tables in S1 File. To assess the alignment of these manikins with the general population, we measured their facial dimensions—interpupillary distance, bizygomatic breadth, lower face height, and ear-sellion depth—on manikins we 3D printed in nylon and assembled with a skin-like silicone layer (methodology in S8 Text in S1 File). We plotted these measurements against the U.S. average data for respective age groups (Fig 3) and found them to fall within the average range for the pediatric population with the maximum deviation from the average ranging in between -3.5/+2.1% across the 5–14-year-old headforms (S6 Table in S1 File).

To create a headform representing a 2-year-old child (named Betty), we reduced the size of the Dizzy headform to 88.5% (S6 Table in S1 File) as we were not able to locate an equivalent headform for that age in any publicly available reference. Dizzy was used for the scaling down (S8 Table in S1 File) as it appeared to be a better representative of the average of various facial measurements compared to Roberta (S7 Table in S1 File), based on the facial measurements of the U.S. average.

Because pediatric facemask have ear loops/straps (and go behind ears) we incorporated a thinner 1mm thick skin-mimicking silicone region where the masks loops around the ears [20]. Step by step description of the fit measurement protocol is provided in supporting information (S9 Text in S1 File). A TSI Model 8048 Portacount Respirator Fit Tester which is used in quantitative fit testing in adults was used with the pediatric headforms. Fit-factor was measured for each flow rate individually by measuring the concentration of sodium chloride

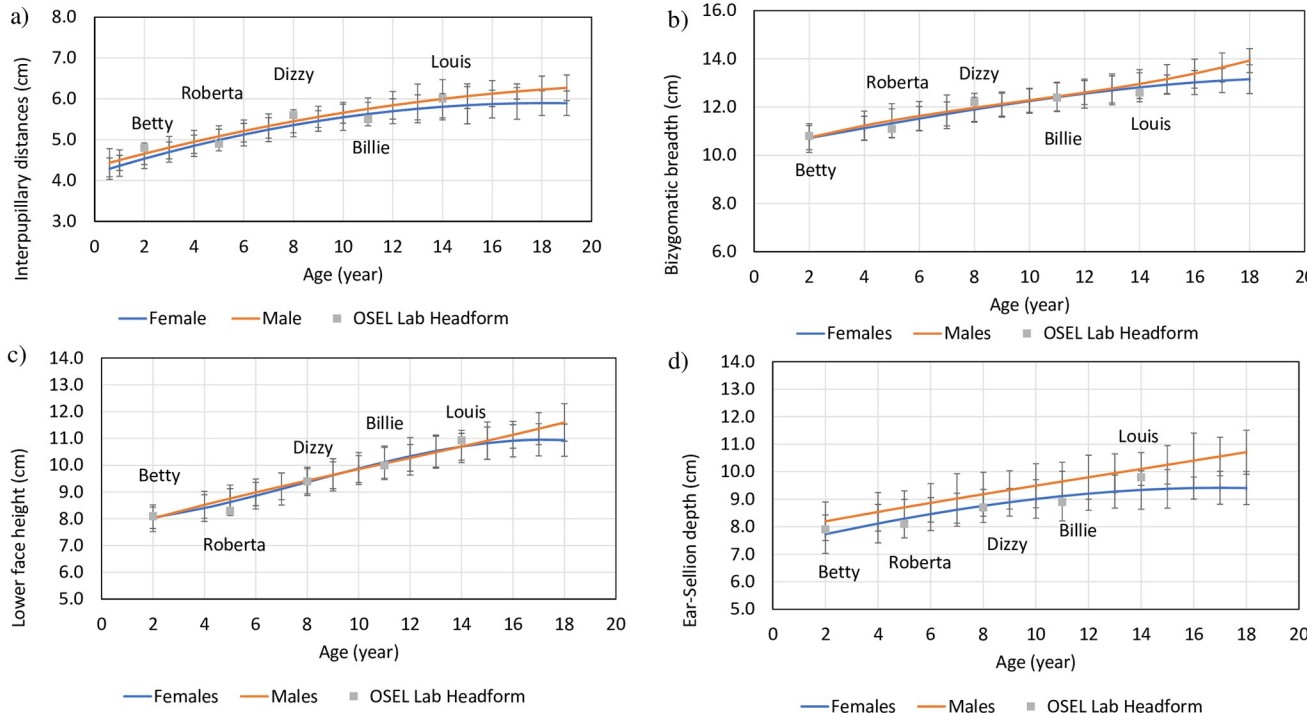

**Fig 3. Facial Dimensions: a) Interpupillary Distance, b) Bizygomatic Breadth, c) Lower Face Height, and d) Ear-sellion Depth of the general population and the headforms employed in this study.** The blue and orange lines were constructed using average values based on literature. The error bars were obtained from values reported in literature (S4 Text in S1 File). The legends (grey square boxes) represent the following headforms that were printed in our laboratory. Betty: 2 year old; Roberta: 5 year old; Dizzy: 8 Year old, Billie: 11 year old; Louis: 14 year old. More details about the headforms chosen appear in S6 Table in S1 File.

(generated using a TSI Model 3026) before ($C_{in}$) or after ($C_{out}$) the pediatric facemasks.

$$Fit\ factor\ (FF) = C_{out}/C_{in} \tag{3}$$

However, since children may engage in a variety of activities (moderate and heavy) while donning masks, an overall fit factor was determined across three flow rates 5, 30 and 45 LPM using the following equation, where FF1, FF2 and FF3 are the fit factors at 5, 30 and 45 LPM, respectively.

$$Overall\ FF = \frac{3}{\frac{1}{FF1} + \frac{1}{FF2} + \frac{1}{FF3}} \tag{4}$$

Airflow sampling was controlled by using mass flow controller (Alicat, Model # MCR-100SLPM-D) to maintain constant suction flow. To simulate human breathing and compare fit-testing results between constant suction flow and oscillatory flow, we used a QuickLung® Breather breathing simulator (Ingmar Medical) (example flow profiles are provided in S1 Fig in S3 Text provided in S1 File).

## Results and discussion

### Worst-case flow rate and pressure drop

Literature review (S1 Text in S1 File) revealed that a flow rate of 45–60 LPM encapsulates the upper limit experienced during vigorous physical activities in individuals under 18 years old.

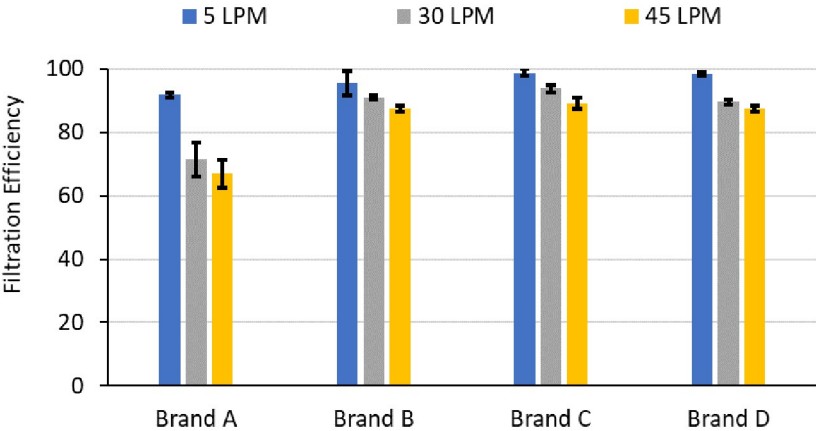

**Fig 4. Filtration efficiency of four pediatric facemask brands at flow rates of 5, 30, and 45 LPM.** The arrangement of the brands reflects their production years, from oldest to newest. Standard deviations shown are based on measurements made in triplicates.

Since pediatric facemasks are indicated for a wide age range of 4–12 years where much lower flow rates are also likely, hence 45 LPM was chosen as a realistic worst-case scenario for evaluating pediatric face masks [14, 21–23].

The inspiratory resistance for adult respirators above which there can be discomfort from donning is 9 mmH$_2$0 at 85 LPM [8]. Independently, based on S5 Table in S1 File, the lowest inspirator resistance reported in children is 0.37 mmH$_2$O/LPM [12], and children's breathing flow rate during vigorous activities can reach 45 LPM. Using Eq (2) and assuming that the total pressure drop in adult lungs ~ 750 Pa (or = 76.5 mmH$_2$O) at 105 LPM [24], and that it remains relatively unchanged at 85 LPM, the maximum permissible pressure drop in a pediatric face mask = 0.37 mmH$_2$O/LPM × 45 LPM × 9.0/76.5 mmH$_2$O = 2.04 mmH$_2$O or ~ 2 mmH$_2$O.

## Filtration efficiency in FDA cleared pediatric facemasks

Fig 4 displays the filtration efficiency of different pediatric facemask brands. On average, findings across four brands indicated consistently high filtration efficiency at low flow rates (96% at 5 LPM), decreasing as flow rates increased (83% at 45 LPM). Brand A demonstrates the steepest (27%) reduction in filtration efficiency when flow rate increased from 5 to 45 LPM, while other brands experienced a lesser (~ 10%) reduction. The variation in filtration efficiency levels obtained for various brands of pediatric facemasks tested in our study aligns with filtration efficiency results reported previously [24–26]. The brands of the pediatric facemasks in Fig 4 are categorized in chronological order of 510(k) clearance received from US FDA, revealing that later generations of pediatric facemasks exhibit higher filtration efficiency than their predecessors, indicating an improvement in subsequent facemask filtration performance.

## Pressure drop in FDA cleared pediatric facemasks

Note that the pressure drop observed for the whole mask coupons is not the pressure that a user will experience as the pressure drop experienced by a user (also referred to as breathability or breathing resistance) is also a function of how well the mask is donned to the user's face. The pressure drop measured on whole masks, nevertheless, provides an important insight as it enables a comparison across brands in a situation where there is no leakage. Fig 5 illustrates

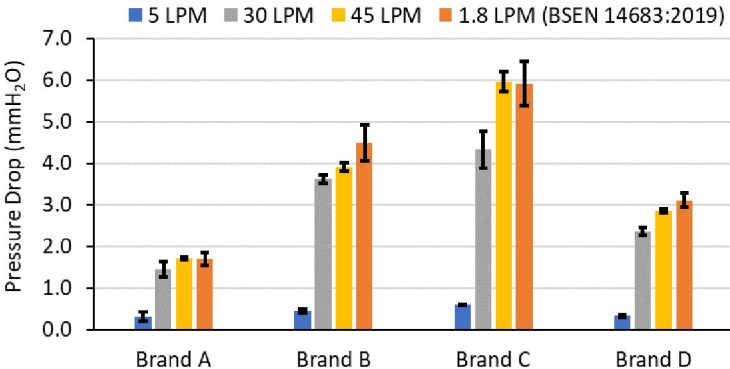

**Fig 5. Pressure drop values for four pediatric facemask brands at flow rates of 5, 30, and 45 LPM, utilizing the experimental setup illustrated in Fig 1a.** To assess pressure drop in the coupons at 1.8 LPM, we employed a technique previously described in the literature, BSEN 14683:2019 [17]. Standard deviations shown are based on measurements made in triplicates.

pressure drop values for four pediatric facemask brands at flow rates of 5, 30, and 45 LPM, utilizing the whole masks with pleats opened and with experimental setup depicted in Fig 1a. Keeping the surface area of the masks the same (approximately 85 cm$^2$), and consistent with previous studies [3], increasing flow rate from 5 LPM to 45 LPM shows an increase in the pressure drop (Fig 5) for all brands of pediatric facemasks. However, there is large brand-to-brand variability. Brand C, for instance, experiences a three-fold higher pressure drop than brand A at 30 LPM and a similar 3.5-fold difference at a higher flow rate of 45 LPM. Liu et al. studied the relationship between filtration efficiency and pressure drop [27]. Consistent with that study, facemask brand C exhibited the highest filtration efficiency (Fig 4) among the four types of masks, accompanied by the highest pressure drop (Fig 5).

When using the BSEN 14683:2019 (Fig 1b) [17] method, the pressure drop is measured on a much smaller un-pleated coupon area (4.9 cm$^2$ compared to ~ 85 cm$^2$). Although, the flow rate is significantly lower, the face velocity for BSEN 14683:2019 (= 1.8 LPM/4.9 cm$^2$ = 6 cm/s) is comparable to the face velocity of facemasks with the pleats opened at the highest flow rate leading to similar pressure drops (since pressure drop linearly scales with velocity). Given the similarities in results reported with whole masks at 45 LPM and with coupons using BSEN 14683:2019, it may be simpler to use the BSEN 14683:2019 method at 1.8 LPM instead of measuring pressure drop of whole masks at multiple flow rates.

### Donning pediatric facemasks: Impact of the pleats

While users are expected to open the pleats when donning a pediatric facemask, lack of instructions may inadvertently prompt children to wear the masks without opening the pleats. To understand the impact of unopened pleats on pressure drop, we first measured the pressure drop in all 4 brands of masks with the pleats opened, following which we then remeasured with the pleats unopened. Unopened pleats resulted in a clear increase in pressure drop with brands B and C (Fig 6a) showing a pressure drop increase that exceeded 25 mmH$_2$O at 30 LPM.

In addition, counterintuitively, using a facemask with pleats instead of offering additional filtration, also resulted in a decrease in filtration efficiency (Fig 6b). For instance, facemask brand A and C with open pleats at a flow rate of 30 LPM exhibited a filtration efficiency of 71% and 94%, respectively, which reduced (p value = 0.038) to 63% and 86%, respectively. The

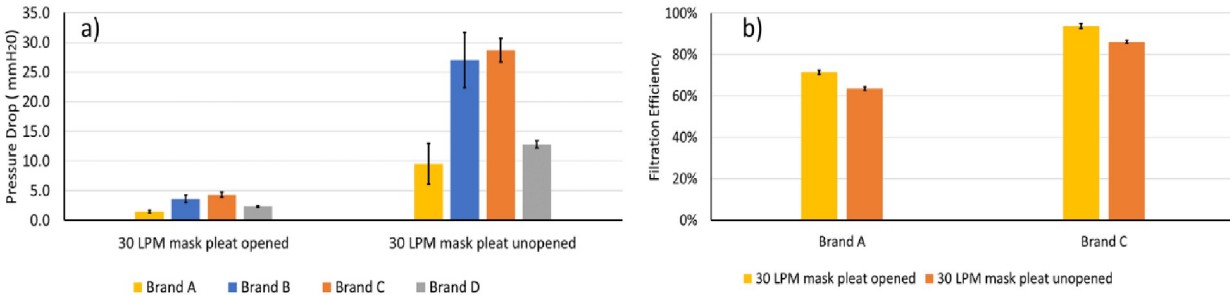

**Fig 6. Pressure drop with open and unopened pleats at 30 LPM for Brands A to D (a), and the impact of filtration efficiency on opened versus unopened pleats (b) for Brands A and C.** The filtration efficiency of Brands B and D with pleats open are shown in Fig 4, while filtration efficiency with pleats unopened were not measured for these two brands. Standard deviations shown are based on measurements made in triplicates.

unopened pleats resulted in about 18% less surface area, and since the face velocity increased in the facemask with pleats (for the same flow rate), and filtration efficiency reduces with increase in face velocity hence it resulted in a ~ 8–9% decrease in filtration efficiency for both brands A and C.

The increase in pressure drop and reduction of filtration efficiency implies that when a child dons a pediatric facemasks with pleats unopened, these facemasks will likely be difficult to breathe through and would cause increased leakage of unfiltered aerosols resulting in poor protection to the user. *The above results underscore the importance of opening the pleats before donning a pediatric facemask to reduce pressure drop, as well as to achieve maximum filtration through these facemasks.*

## Lot to lot variability in facemasks

To investigate potential variations in filtration efficiency and pressure drop among different lots of pediatric facemasks for the same brand, we conducted a limited lot-to-lot comparison involving brand A and C at flow rates of 5 and 30 LPM and found these were similar across two lots with no statistically significant differences found based on the student's t-test ($p > 0.05$) (S4a and S4b Fig in S1 File).

## Fit test and breathing resistance

While the filtration efficiency and pressure drop measurements are important for thorough characterization of a pediatric facemask. Given the lack of clinical studies conducted with pediatric facemasks, it is challenging to derive any meaningful clinical implications based on 10–20% difference in filtration efficiency or 2–3 fold difference in pressure drop across various brands of pediatric facemasks. However, fit factor is a more meaningful metric as a fit factor of 2 versus 8 implies that a person wearing a mask with a higher fit is able to block out significantly greater number of aerosols (by a factor of 4) thus reducing risk of airborne-infection [28]. Therefore, the subsequent sections delineate the brand-to-brand performances using the metric of fit-factor.

**Intended use—Fit factor measurements of child headforms across various brands of pediatric facemasks.** In Fig 7a, the results of the fit test on the 8-year-old Dizzy headform are illustrated. Brands A, B and D show higher fit factor at low flow rates that gradually declines with increasing flow rate which is representative of heavier activities that result in increased exertion during breathing and or increase of age (as inhalation flow rate increases with age).

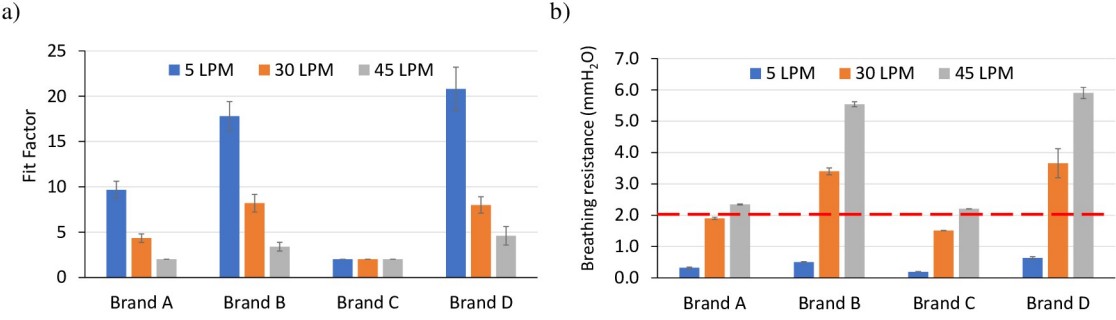

**Fig 7. Results of the fit test: a) fit factor, b) breathing resistance measurements for four brands of pediatric facemasks worn on the 8-year-old Dizzy headform.** Standard deviations shown are based on measurements made in triplicates.

Curiously, Brand C, despite its higher filtration efficiency (Fig 4), consistently demonstrated a lower fit factor compared to other brands across all flow rates, potentially due to poorly deformable nosepieces that did not conform to the contour of the manikin nose leading to leaks. In contrast, brand A exhibits a fit factor 5 times higher than brand C at lower flow rates, which gradually decreases as the flow rate increases. Similarly, brand B and D displayed fit factors 8 and 10 times higher than brand C at lower flow rates, respectively. The observed trends are similar for the 11-year-old Billie and 5-year-old Roberta headforms and is reported in the form of overall fit-factor in the next section (Fig 8a).

Fig 7b illustrates breathing resistance measurements for four pediatric facemask brands on the 8-year-old Dizzy headform. As flow rates increase, breathing resistance increases across all brands, indicating difficulty in breathing at higher flows. The red dotted line indicates the 2 $mmH_2O$ that we determined by extrapolating from evidence on adult respirators. Majority of the brands exceed this 2 $mmH_2O$ threshold at the higher flow rates of 30–45 LPM. This suggests that children may experience more discomfort and breathing challenges when wearing masks during high-intensity activities (sports) or if they are older (as older age is likely to be associated with higher inspiratory flow rate).

To investigate potential variations in fit and breathing resistance among pediatric facemasks from the same brand but different lots, a lot-to-lot comparison was conducted. S4c and S4d Fig in S6 Text of S1 File illustrate the overall fit factor and breathing resistance values for brands A and C at flow rates of 30 and 45 LPM. Both plots reveal no significant differences, as determined by Student's t-test ($p > 0.05$) implying that our findings are likely valid across multiple lots of pediatric facemasks.

To simulate realistic pediatric population breathing and compare fit-testing results between constant suction flow and oscillatory flow, we conducted fit tests on four brands of pediatric facemasks across three pediatric headforms. S5a Fig in S7 Text of S1 File illustrates the overall fit factor values for on the 8-year-old Dizzy headform using oscillatory and constant flow rates. On average, the overall fit factor for brands A, B, C, and D was 3, 6, 2, and 8, respectively. These values were found to be similar for oscillatory and constant flow rates based on Student's t-test ($p > 0.05$) across all three headforms (S5a-S5c Fig in S1 File) implying that a simpler constant suction flow rate test set up may be a good representative set up to measure overall fit factors.

**Off label use—Manikin fit measurements when used for younger or older children.**
The overall fit and highest breathing resistance in headforms of 5, 8, and 11-year-olds, following the recommended age range of 4 to 12 years for pediatric facemask usage is shown in Fig 8a and 8b. This situation constitutes an intended use scenario as the pediatric facemasks FDA

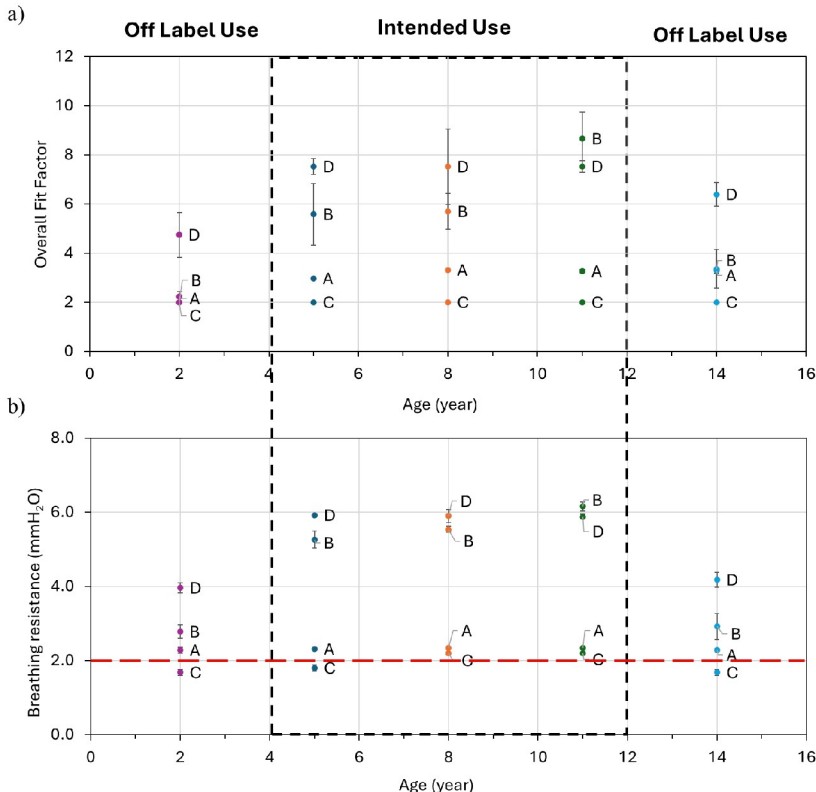

**Fig 8. a) Overall fit factor and b) breathing resistance of pediatric facemasks (brands A, B, C, and D) across all head forms.** Standard deviations shown are based on measurements made in triplicates.

has cleared so far are for that specific intended population. We also explored mask performance on headforms representing ages outside this typical range, specifically 2- and 14-year-olds (Fig 8a and 8b) which we refer here as "off label" as these FDA cleared facemasks are not intended to be used for this younger and older age groups.

On average, we observed a 43% decrease in the overall fit factor and a 33% decrease in breathing resistance for the 2-year-old headform compared to intended use scenarios. Similarly, a 22% decrease in the overall fit factor and a 31% decrease in breathing resistance were noted for the 14-year-old headform. This decline is attributed to pediatric facemasks not designed for children younger than 4 and older than 12, leading to inadequate face coverage, which concomitantly caused leaks and decreased overall quantitative fit factor and lowered breathing resistance. Given our findings of reduced overall fit in off-label situations, it underscores the need for developing pediatric facemasks for < 4 year and > 12-year age groups. This is particularly important as CDC recommends masks to be worn by children older than 2 years of age.

An alternative for the older pediatric population (> 14 and above) would be use of N95 respirators which are indicated for use in workplaces and by adults. Although our findings for overall N95 respirator fit for the 14-year-old manikin was found to be very high (quantitative fit factor of 200, S12a Fig in S1 File), the significant breathing resistance of N95 respirators relative to pediatric facemasks (S12b Fig in S1 File) would likely hinder the practicality of this approach. It may be beneficial for the academic and the medical device community to engage in more research on developing respirator designs that may be suitable for older children,

providing high quantitative fit ($>10$) compared to pediatric facemasks while maintaining low pressure drop and breathing resistance at reasonably high flow rates of 30–45 LPM.

## Practical implications for the community

### General public.

1. Proper Mask Usage Practices: Given the emphasis on opening pleats before donning the mask, it's crucial for the general public, especially parents and caregivers, to be educated on proper mask-wearing practices to ensure optimal breathability and minimized leakage.

2. Age-Appropriate Mask Selection: Parents should pay attention to age recommendations when choosing pediatric facemasks. This study suggests that masks not intended for children older than 12 or younger than 4 may lead to inadequate face coverage implying low quantitative fit.

3. High filtration efficiency may not correlate with good fit: Parents and caregivers should ensure that the nose-clip of the facemask conforms to the child's nose bridge. This will help ensure better fit and maximum protection to wearer. Without this step, even a higher filtration efficiency mask may not offer adequate protection to the wearer.

### Pediatric facemasks (device) manufacturers.

1. Design Optimization for Fit: Manufacturers may ensure that nose clips are designed for optimal fit and the clips can conform to the nose bridge adequately. This will help ensure proper fit and protection to the wearer. It may also be beneficial to develop a test method to assess the malleability of nose-clips and characterize the clips for better fits.

2. Newer Masks for Specific Age Groups: Considering the absence of masks for children under 4 and over 12 years old, there is an opportunity for manufacturers to develop and introduce masks tailored to these age groups. This addresses a current gap and ensures a more comprehensive range of protective options for pediatric populations.

3. Pressure Drop Considerations: There is a need for developing more optimal pediatric facemasks designs with minimal breathing resistance ($\sim 2$ mmH$_2$O) at relatively high flow rates (45 LPM) which is lower than the breathing resistance of $< 5$ mmH$_2$O described in ASTM F3502 for Barrier Face Coverings [29]. However, what breathing resistance may be optimal would likely depend on the age range the mask design is indicated for.

### Academia and future research.

1. Bench top studies: Using 3D-printed child manikins to assess fit across a broader spectrum of diverse anthropometric features.

2. Breathability: Fit-testing performed on children with various brands of facemasks for determining what nominal fit-factor may offer reasonable protection to children and to assess if the brand-to-brand differences in pressure drop across pediatric facemasks are clinically meaningful. When making such assessments it would be important to first fully characterize the mask-brand used for filtration, pressure drop, as well as assess the performance of nose-bridge strips and ear loops. Additionally, development of pediatric facemasks that can be used by those younger than 4 years, or older than 12 years old. As well as development of a stratified optimal pressure drop range for various pediatric age groups including 2–4 years, 4–12 years, as well as those above 12 years of age.

3. Inclusion of Various Ethnicities and Diversity: We did not consider various ethnicities due to data constraints, future research should strive to incorporate a more diverse demographic. This inclusivity would enhance the generalizability of findings and ensure that pediatric facemasks are evaluated across a spectrum of ethnic backgrounds. Assessment of the difficulties around pediatric mask-donning for children who may have developmental challenges or lung diseases (e.g. asthma). These studies should also include assessment of typical flow rates for diseased lungs so the masks designed can be tested at relevant flow rates.

4. Long-Term Wear Effects: Investigating the prolonged use of pediatric facemasks among children could offer insights into the long-term effects, comfort, and potential challenges associated with extended wear. This aspect is particularly relevant in scenarios where continuous mask usage is required, such as in school settings.

5. Activity Levels and User Experience: Exploring the impact of different activity levels on mask performance, and subjective experiences of children would provide valuable insights. Understanding how masks perform during various physical activities can guide the development of masks tailored to the diverse needs of active children, ensuring both protection and comfort. Considering factors such as comfort, breathability, and overall satisfaction, would contribute to increased compliance.

6. Impact of Environmental Conditions: Considering the influence of environmental conditions, such as humidity and temperature, on mask performance would provide valuable information for ensuring effectiveness in various real-world situations.

7. Incorporation of Patient-Specific Factors: Assessing how patient-specific factors, such as respiratory conditions or facial anatomy variations (beards for adolescents, injuries), may influence mask performance is an avenue for future exploration. This personalized approach could contribute to the development of more tailored and effective pediatric facemasks.

## Limitations

Limited Range of Testing Exercises: Our study focused on assessing facemask fit based on specific breathing exercises (normal, deep breathing at 30 and 45 LPM). While these exercises provided valuable insights, it's crucial to note that respirator fit testing typically performed on adults involves a more diverse range of movements, such as head turns, up-and-down motions, and talking [30], and is itself not intended to fully reproduce motions of subjects in a true Workplace Protection Factor study. The absence of these additional movements in our assessment could impact the findings.

Individual Variability: Despite our efforts to cover a spectrum of age groups using different pediatric headforms, human facial features vary widely among individuals. Factors like ethnicity, age, gender, and facial dimensions significantly influence facemask fit and breathing resistance. Our study's reliance on specific headforms may not fully encapsulate this diversity in the pediatric population.

Simplifications in Skin Thickness: Because of lack of information and to simplify our methodology we didn't incorporate variable skin thickness in our headforms. Although our previous research on adults suggested that these simplifications didn't significantly affect the results when compared to adult N95 respirator fit testing, it's essential to recognize that pediatric facial structures might respond differently. The absence of variable skin thickness might

influence the accuracy of our fit-testing measurements. Given the simplifications in our study, our quantitative fit results should be interpreted with caution.

Breathing Simulation: While we did not use breathing simulator extensively, the limited studies we conducted demonstrated similarity between constant and oscillatory flow rates.

Ethnic Diversity: Various ethnicities were not considered due to a lack of data. While not studied in this context, the protocols described can still be used by modifying headform measurements to further our understanding of the impact of various racial ethnicities on fit.

Breathability in Diseased Lungs: What amount of breathing resistance would be tolerable for children with asthma or other conditions was not studied.

## Conclusions

Leveraging our validated adult manikin fit-test method adapted for the pediatric population, we developed a methodology for evaluating the fit factor and breathing resistance of cleared pediatric facemasks. We then assessed four pediatric facemask brands available in the U.S. market, across 2–14 year old pediatric manikins to provide insights on how well pediatric facemasks are likely to perform in real-world situations and future considerations.

Key findings emphasize the necessity of a comprehensive evaluation covering all aspects of pediatric facemasks. Filtration efficiency (in absence of any leaks) was consistently high (>80%) for majority of the brands even at relatively high flow rates of 45 LPM. Brands exhibited substantial differences in pressure drop, with some brands surpassing pressure drop limit of 2 $mmH_2O$ at high flow rates that may lead to discomfort for the child wearer. Fit may not correlate with high filtration efficiency, underscoring the critical role of meticulous design in ensuring optimal fit, particularly for nose-clips. Our findings also highlighted the need for developing facemasks for those below 4 years as well as above 12 years of age.

## Supporting information

**S1 File.**
(DOCX)

## Acknowledgments

The contents of this article do not necessarily represent policy or current thinking of the U.S. Food and Drug Administration or the Department of Health and Human Services. Dr. Ali Hasani was a research fellow at the U.S. Food and Drug Administration via the ORISE program through the Oak Ridge Associated Universities.

## Author Contributions

**Conceptualization:** Dana Rottach, BiFeng Qian, Suvajyoti Guha.

**Data curation:** Ali Hasani, Bryan Ibarra, Suvajyoti Guha.

**Formal analysis:** Ali Hasani, Bryan Ibarra, Suvajyoti Guha.

**Funding acquisition:** Kirstie Snodderly, Suvajyoti Guha.

**Investigation:** Ali Hasani, Bryan Ibarra, Kirstie Snodderly, Suvajyoti Guha.

**Methodology:** Ali Hasani, Kirstie Snodderly, Daniel Porter, Suvajyoti Guha.

**Project administration:** Suvajyoti Guha.

**Resources:** Suvajyoti Guha.

**Supervision:** Dana Rottach, BiFeng Qian, Daniel Porter, Suvajyoti Guha.

**Validation:** Ali Hasani, Bryan Ibarra, Suvajyoti Guha.

**Writing – original draft:** Ali Hasani, Suvajyoti Guha.

**Writing – review & editing:** Bryan Ibarra, Kirstie Snodderly, Dana Rottach, BiFeng Qian, Daniel Porter, Suvajyoti Guha.

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
