## [Decision Letter · Decision Letter 0]

10 Jun 2024

PONE-D-24-14922A Retrospective Characterization of Pediatric Facemasks Marketed in the US and Implications for Future DesignsPLOS ONE

Dear Dr. Guha,

Thank you for submitting your manuscript to PLOS ONE. After careful consideration, we feel that it has merit but does not fully meet PLOS ONE’s publication criteria as it currently stands. Therefore, we invite you to submit a revised version of the manuscript that addresses the points raised during the review process.

The studies performed by the authors are impactful and meaningful as they provide experimental methods and procedures to characterize various performance attributes (including filtration efficiency, fit, and breathing resistance) of facemasks. The manuscript is overall well written, i.e., it is easy to follow, is concise, and cites appropriate literature. The manuscript especially suits PLOS One due to the broad and general nature of this work.

Both the reviewers suggested minor changes that would be helpful to incorporate to improve the manuscript further.

Along with the reviewer’s suggestions, I have two suggestions for the authors-

The authors do not specify the justification behind equations 1 and 2. It is unclear why the ratio of maximum permissible pressure drop to total pressure drop should be the same for adults and pediatric populations.It is unclear whether differences suggested by the authors between different brands in terms of breathing resistance or filtration efficiency are practically meaningful. I would encourage authors to add a discussion on what differences in these parameters are practically relevant and will be impactful. Given that certain brand facemasks perform better than others in various experimental tests performed by the authors, can authors cite particular anecdotes suggesting the practical implications of these differences?

We look forward to receiving your revised manuscript.

Kind regards,

Harshit Agarwal

Guest Editor

PLOS ONE

“COVID-19 Research Funding”

“None to report”

6. Please amend either the title on the online submission form (via Edit Submission) or the title in the manuscript so that they are identical.

Reviewers' comments:

Reviewer's Responses to Questions

**Comments to the Author**

1. Is the manuscript technically sound, and do the data support the conclusions?

Reviewer #1: Yes

Reviewer #2: Yes

2. Has the statistical analysis been performed appropriately and rigorously? 

Reviewer #1: I Don't Know

Reviewer #2: Yes

3. Have the authors made all data underlying the findings in their manuscript fully available?

Reviewer #1: Yes

Reviewer #2: Yes

4. Is the manuscript presented in an intelligible fashion and written in standard English?

Reviewer #1: Yes

Reviewer #2: Yes

5. Review Comments to the Author

Reviewer #1: This manuscript is a good explanation of scientific research. I want to write some suggestions important for polishing the manuscript.

1. The abstract needs to mention the future prospects of your studies. It would be helpful to add brief statements about practical utility, limitations, and proposed future work of this study to the abstract.

2. Figure 1 (b) contains more subdivisions. Please make sure that each subdivision (A, B, and C) is explained in the figure caption.

Reviewer #2: In the manuscript titled, “A Retrospective Characterization of Pediatric Facemasks Marketed in the United States and Implications for Future Designs”, the authors presented three major objectives of the study which is to determine a suitable flow rate for the testing of the pediatric facemasks, define an acceptable pressure drop for the pediatric facemasks, and assess fit and breathing resistance. Even with the limitations of this study presented by the authors as well (such as the need to study pediatric population with respiratory illness and asthma, and other limitations) and some future suggestions in section 4 and 5, the study presented a good starting point to provide a workflow to evaluate the pediatric facemasks and present the need to manufacture facemasks specifically designed for various sub population that is below 4 years and above 12 years of age, which in the future might need a more detailed quantitative cutoff of the various design aspect of the pediatric facemask such as suitable flow rate, pressure drop, fit and breathing resistance.

I have the following minor suggestions and questions.

1) In abstract, results & discussion on page 9, line 3, “retrospective assessment of four brands of legally marketed facemasks in the U.S. …” does not clarify that there are only four brands available in the U.S. for pediatric masks as mentioned later in the paper. It would be great if the authors could clarify there are only four brand available in the abstract itself. Further in the same line, “…revealed that majority of the brands have high filtration efficiency (>95%) at low flow rate 5 LPM reduces to ~ 80% at 45 LPM.” could be edited to “…5 LPM which reduces to …” with the addition of ‘which’ or a similar change.

2) In the abstract, results & discussion on page 10, line 1, “Furthermore, opening the pleats of the facemasks completely results in a notable reduction in pressure drop (a 6.6-fold decrease, p=0.03).”, the authors could also add a line about the point of how the opening of pleats affected the efficiency which is later discussed in the results section.

3) In section 2 d) Measuring Filtration Efficiency and Pressure Drop on page 13, line 3, a definition of facemask coupons would help.

4) In section 2 e) Method for Measuring Fit and Breathing Resistance on page 15, line 4 the authors mention the selected headforms represented pediatric individuals aged 2 to 14 years. If the age considered from 2 to 14 years is to study the off-label use as mentioned later than it would be good to clarify why the authors study 2 to 14 years here. Also, considering the use of the pediatric masks only for the FDA approved age group which is 4 to 12 years according to the authors, and considering the facial dimensions of the Roberta, Dizzy, and Billie from Figure 3, the three individuals might represent only part of the spread of general pediatric population facial features for example the Figure 3 a) suggests that Roberta and Billie are right below the average values for interpupillary distance and only Dizzy was above average, while for Figure 3 d) all three were right below average. All the three data points still fit within the deviation from the averages but for a study determining the pressure drop, and breathing resistance limits along with considering fits, a better representation of the pediatric population might be with the additional data points for the lowest possible values of facial dimensions at the lower age limit and highest possible facial dimensions at the highest age limit.

5) In supplementary section figure S2 the labels are overlapping.

6) In section 2 e) Method for Measuring Fit and Breathing Resistance on page 16, line 9 what are the acceptable limits of fit factor to be considered a good fit. It is later suggested on page 25 that > 10 might be a high quantitative fit but some quantitative measure while defining the fit on page 16 would be better.

7) In section 3, page 17 line 4, “hence 45 LPM was chosen as a realistic worst-case scenario for evaluating pediatric face masks [14, 21-23].” What is the cutoff used to define ‘realistic worse-case scenario’ compared to the actual maximum flow rate. Also, a clarification as to how the realistic value for worse case scenario might be affected based on considering the limitations discussed later such as respiratory illness with current age group of 4 to 12 years, or future design for use in population with >14 years of age, would be great if added to the discussion.

8) In section 3, page 17 line 10, consider changing “~ 750 Pa or (= 76.5 mmH2O)” to “~ 750 Pa (or = 76.5 mmH2O)” with the ‘or’ within the brackets.

9) In section 3, page 17 line 11, is “0.37 mmH2O/LPM × 45 LPM” considered as defined on page 13 equation 1. If yes, it would be better to clarify that.

10) In section 4 b) 1. on page 25, the authors state, “Manufacturers may ensure that nose clips are designed for optimal fit and the clips can conform to the nose bridge adequately. This will help ensure proper fit and protection to the wearer.” It would be great to determine a quantitative measure for how ductile or malleable or a similar measure of the material of the nose clip would be required to allow a better fit as the fit/efficiency highly depended on the fit of the nose clip (specifically for brand C).

11) In section 4 b) 3. on page 25, the authors provide quantitative measures for the manufacturers, “There is a need for developing more optimal pediatric facemasks designs with minimal breathing resistance (~ 2 mmH2O) at relatively high flow rates (45 LPM) which is lower than the breathing resistance of < 5 mmH2O described in ASTM F3502 for Barrier Face Coverings [28].” Would the 45 LPM upper limit of the flow rate to be tested change based on the age group considered as suggested in section 4 b) 2.? If yes, it would be better to clarify the age group or mention it could depend on the age group in section 4 b) 3.

12) Figure 4 to 8 need a statement defining what the error bars represent in the figures.

6. PLOS authors have the option to publish the peer review history of their article (what does this mean?). If published, this will include your full peer review and any attached files.

Reviewer #1: **Yes**

Reviewer #2: No

---

## [Author Response · Author response to Decision Letter 0]

8 Jul 2024

We would like to thank the editor and the reviewers for their time. Please refer to attached document "response to reviewers" for detailed point by point rebuttal.

---

## [Editor Report · Decision Letter 1]

15 Jul 2024

A retrospective characterization of pediatric facemasks marketed in the United States and implications for future designs

PONE-D-24-14922R1

Dear Dr. Guha,

We’re pleased to inform you that your manuscript has been judged scientifically suitable for publication and will be formally accepted for publication once it meets all outstanding technical requirements.

Within one week, you’ll receive an e-mail detailing the required amendments. When these have been addressed, you’ll receive a formal acceptance letter, and your manuscript will be scheduled for publication.

Kind regards,

Harshit Agarwal

Guest Editor

PLOS ONE

Additional Editor Comments:

I thank the authors for submitting the manuscript to PLOS ONE and making appropriate changes to the manuscript based on the comments from reviewers and the editor. The manuscript is clear and concise and offers novel methods, experimental approaches, and results related to design, characterization, and use of pediatric facemasks in US. I believe the manuscript to be of value and interest to the PLOS ONE readership.

---

## [Editor Report · Acceptance letter]

10 Sep 2024

PONE-D-24-14922R1 

PLOS ONE

Dear Dr. Guha, 

I'm pleased to inform you that your manuscript has been deemed suitable for publication in PLOS ONE. Congratulations! Your manuscript is now being handed over to our production team.

Kind regards, 

on behalf of

Dr. Harshit Agarwal 

Guest Editor

PLOS ONE